# Comparative Assessment of Phenolic Profiles, Cellular Antioxidant and Antiproliferative Activities in Ten Varieties of Sweet Potato (*Ipomoea Batatas*) Storage Roots

**DOI:** 10.3390/molecules24244476

**Published:** 2019-12-06

**Authors:** Yiming Sun, Zhijun Pan, Chunxian Yang, Zhenzhen Jia, Xinbo Guo

**Affiliations:** 1Key Laboratory of Eco-environments in Three Gorges Reservoir Region (Ministry of Education), School of Life Sciences, Southwest University, Chongqing 400715, China; s8355@swu.edu.cn (Y.S.); yangchx@swu.edu.cn (C.Y.); 2School of Food Science and Engineering, South China University of Technology, Guangzhou 510640, China; 201664428579@mail.scut.edu.cn; 3Research Center of Agriculture, Pingdingshan Academy of Agricultural Science, Pingdingshan 467000, China; zhenzhenjia_001@126.com

**Keywords:** sweet potato, phytochemicals, cellular antioxidant activity, antiproliferation activity, cytotoxicity

## Abstract

Sweet potato is the sixth most important crop widely cultivated around the world with abundant varieties. Different varieties gain different phenolic profiles which has drawn researchers’ attention for its unique health benefits. Our study evaluated the phenolic profiles, total and cellular antioxidant activities, antiproliferative activities, and cytotoxicity in 10 cultivated varieties of sweet potato in different colours. Among fourteen metabolites detected in our study, hyperoside, ferulic acid and caffeic acid were considered as prominent in SPSRs. According to the principle component analysis, phytochemical composition of HX22, YS15 and YS7 was quite similar. The results also evidenced that purple-fleshed varieties, such as YS43, YZ7 and YY153, have higher total phenolics content and corresponding stronger total antioxidant capacities as well as cellular antiproliferative activities against human liver cancer HepG2 cells than other varieties. The extremely significant correlation between phenolics and total antioxidant activity was also revealed by Pearson correlation analysis (*p* < 0.05). However, no significant relevance was found between intracellular antioxidant activity and total phenolic content or flesh colour of sweet potatoes.

## 1. Introduction

Sweet potato (*Ipomoea batatas (L.) Lam.*, *family Convolvulaceae*) is an important starch and vegetable crop and widely cultivated in the world, which produces storage roots and edible leaves for human diet. Currently, sweet potato ranks the sixth most important crop after rice, wheat, potato, maize and cassava, which produces more than 105 million tons each year in the world and 95% in developing countries [1]. Many parts of sweet potato are edible, including roots, vines and leaves. Storage roots of sweet potato are abundant of starch, protein, dietary fibre and micronutrients (including vitamins, minerals as well as bioactive compounds), which can provide enough energy and nutrients for health benefits [2]. The skin colours of storage roots vary from white to yellow-orange and dark purple, which contain different bioactive compounds (such as phenolics, flavonoids, anthocyanins and carotenoids) [3,4,5].

Phenolics are secondary metabolites whose synthesis and accumulation is generally stimulated in response to biotic or abiotic stresses in higher plants, which exhibit a wide range of biological effects eliciting health benefits. Previous studies demonstrated that phenolics have potent anticancer activity with different regulation mechanisms including antioxidant, antiproliferative activities and induction of apoptosis [6]. Plant-based foods, such as fruits, vegetables, potatoes, grains and legumes, contain phenolic compounds which have been shown to play a role in the prevention of chronic diseases and to provide desirable health benefits. Epidemiological studies suggested that regular consumption of plant-based foods had a crucial role in the prevention of chronic diseases, such as cancer, diabetes, cardiovascular diseases and neurodegenerative diseases [7].

Sweet potato is considered low on the glycaemic index scale and categorized as one of the potent functional foods because of its high dietary fibre and phenolic content. Previous research focused on bioactive carbohydrates, protein, lipid, carotenoid, phenolic, anthocyanin and mineral composition in different parts or processing of sweet potato [2]. The aim of this research was to evaluate the phenolic profiles, total and cellular antioxidant activities, antiproliferative activities and cytotoxicity in 10 cultivated varieties of sweet potato in different colour from white to yellow-orange to dark purple (Figure 1). Although many studies have been reported that sweet potato has antioxidant activity, there are still a paucity of information about the phytochemical profiles and bioactivities in sweet potato. Whether antioxidant capacity is related to human health is still controversial. However, we hypothesise that antioxidant capacity could be an implication or indicator for human diet and health. Epidemiological and clinical studies show that diets with a high antioxidant capacity are related to significant decreases in plasma C-reactive protein, ischemic stroke or the overall risk of cardiovascular disease and colorectal cancer [8,9,10]. Previous studies have only disclosed the antiproliferative or anticancer activities of SPSRs extracts against NB4, MCF-7, HCT-116, HeLa, SNU-1, WiDr and MDBK cell lines, indicating SPSRs might have potential anticancer benefits [11,12,13,14]. To the best of our knowledge, we were the first one who gave the antiproliferation test of SPSRs extracts to HepG2 cell line, providing more information in this area.

Further information about the phytochemical composition and antiproliferation activity of SPSRs can provide a breeding strategy for bioactive compounds fortification in sweet potato as a functional crop and value-added food products to promote human health. Such research should be proved useful in the determination of the bioactivity to justify a broader use of sweet potato in health-related applications. Additionally, measuring the antioxidant capacity of SPSRs might provide more information for designing a healthier diet from the perspective of antioxidant capacity state.

## 2. Results

### 2.1. Total Phenolic Content in Different Varieties

Figure 2 clarified the total phenolic content in ten selected varieties of sweet potato storage roots (SPSRs) ranging from 30.73 ± 1.25 mg GAE/100 g FW to 492.89 ± 17.13 mg GAE/100 g FW, average at 151.34 mg GAE/100 g FW. The purple-flesh varieties YS43 was the variety with highest total phenolic content which was more than sixteen times as much as the lowest ones YS7. The small fluctuation ranging from 0.49 to 17.13 mg GAE/100 g FW commendably ensured the stability and reliability of the results given by our study.

### 2.2. Phytochemical Composition of Ten Varieties

A total of fourteen phytochemical compounds were detected in our study, including flavonoids and phenolic acid, shown in Table 1 with RP-HPLC method. Additionally, a principle component analysis was also carried out and showed in Figure 3. The closer the sample points were, the more similar their composition was. The vectorial angles among different compounds indexed the relevance among those substances.

Caffeic acid was detected in all varieties except YS7, ranging from 0.13 ± 0.01 mg/100 g FW to 4.57 ± 1.18 mg/100 g FW with small fluctuation among varieties. In addition to two purple-fleshed varieties, YS43 and YZ7, ferulic acid was calculated in other eight varieties with the range of 1.31 ± 0.003 ~ 27.97 ± 0.13 mg/100 g FW. As for the group of flavonoid, content of hyperoside was fairly high in YS43 and YY153 and could be detected in most varieties in our study at the range of 7.94 ± 0.97 to 471.3 ± 2.5 mg/100 g FW. The difference of hyperoside content among different varieties was significant. The highest content of hyperoside was proximately sixty times as much as the lowest one. Therefore, we presumed that ferulic acid and caffeic acid could be the prominent phenolic acids and hyperoside was the prominent flavonoid in SPSRs.

According to Figure 3, YS15, HX22 as well as YS7, which have the lightest colour among ten varieties, had similar values, indicating they had similar phytochemical composition in our study. However, the high quantity of hyperoside distinguished YS43 and YY153 which were both purple-fleshed varieties from others including another purple-fleshed one YZ7. The vector of epicatechin, isorhamnetin, isoquercetin and syringic acid kept closely to each other in Figure 3, indicating a high relevance among them. However, hyperoside showed no significant association among those four compounds mentioned above because the vector of hyperoside was nearly vertical to those four vectors. Additionally, catechin and chlorogenic acid resisted to other compounds because they received an opposite vector direction toward those four compounds. Rutin hydrate, trans-ferulic acid, catechin and chlorogenic acid also showed high correlation between each other.

### 2.3. Total Antioxidant Activity

The total antioxidant activities of different varieties were measured by ORAC assay and are shown in Table 2. ORAC is a sensitive high-throughput assay that can accurately reflect the oxygen radical absorbing activity. The results showed that CAA values among those ten varieties ranged from 0.40 ± 0.03 to 7.38 ± 0.98 mmol TE/100 g FW. The darkest variety YS43 performed the best antioxidant activity among ten varieties which was 18.5 times as much as the lowest one CS1.

### 2.4. Cellular Antioxidant Activity

Another assay was also used for estimating the antioxidant activity in HepG2 cells in this study. With a PBS wash protocol treated by samples, this method tended to represent the complexity of biosystem including the cellular absorption, metabolism and distribution of antioxidant. The CAA value with a PBS wash could be consider as an indicator of the intracellular antioxidant activities [15]; its helpful for making a good prediction of antioxidant activity in vivo and evaluating whether phytochemicals could easily across the cell membrane or not.

To ensure the accuracy of the results, the extracts were tested at concentrations lower than CC10, avoiding the impact of cellular cytotoxicity. The results of cytotoxicity test against HepG2 cells are presented in Figure 4B, and the CC10 values ranged from 0.698 mg/mL to 5.304 mg/mL. The results were placed in Table 2 as well.

Based on Table 2, all CAA values were lower than total antioxidant values calculated by ORAC assay. The CAA values with PBS wash detected in our study ranged from 0.46 ± 0.08 to 181.15 ± 18.24 μmol QE/100 g FW. Except for YS15, all the phytochemical extracts could restrain the increasing emission of fluorescence due to DCFH. In the PBS wash protocol, XY34, YS43 and YZ7 had higher CAA values of 181.15 ± 18.24, 120.68 ± 36.9, 50.73 ± 7.56 μmol QE/100 g FW, respectively, followed by WS7, YY153, HX22, YS25, YS7, CS1 and YS15.

### 2.5. Antiproliferative Activity against Human Liver Cancer HepG2 Cell

The dose-dependent effect of all extracts on the growth of HepG2 cell line in vitro was demonstrated in Figure 4A and presented as IC50 ranging from 4.663 mg/mL to 351 mg/mL. A lower IC50 value indicated a stronger antiproliferative activity. Except for the lightest flesh colour sample, YS15, which barely inhibited HepG2 cell proliferation, all the other samples revealed a positive correlation between the extract concentration and antiproliferative activity. Among those samples, YS43, YS7 and YZ7 appeared to show the strongest antiproliferative activity with IC50 values of 4.663 mg/mL FW, 5.162 mg/mL FW and 5.287 mg/mL FW respectively, followed by WS7, HX22, YS25, YY153, XY34, CS1 and YS15.

The extracts of ten varieties were also subjected to the cellular cytotoxicity test to gather a more comprehensive awareness of the inhibition effect of the phytochemical extracts on HepG2 cell. The results were presented in Figure 4B. Within the experimental concentration range, XY34, YS15 and CS1 barely showed inhibition impact on cell proliferation which made a difficulty in estimating CC50 values accurately. Though the extract of YS7 had the strongest cytotoxicity with CC50 value of 6.284 mg/mL and YY153 presented the highest CC50 value of 47.74 mg/mL FW. In the case of extracts from other five varieties, they showed similar cytotoxicity at the range of 14.96 to 22.3 mg/mL.

### 2.6. Correlation Analysis among Phytochemicals and Antioxidant and Proliferative Activities

To further investigate the relationship among phytochemical composition, antioxidant and antiproliferative activities, a Pearson correlation analysis (two-tailed) was carried out and given in Figure 5. ORAC value was found significantly correlated to total phenolics content (*p* < 0.05). Besides, ORAC value was also found positively relative to certain kinds of phytochemical compounds detected in our study, including caffeic acid, trans-ferulic acid and rutin hydrate. Yet, ferulic acid was found inversely related to ORAC value. Surprisingly, no distinct relevance was detected among CAA values in both PBS wash protocol and no PBS wash protocol, ORAC values and total phenolics content.

## 3. Discussion

### 3.1. Comparison of Total Phenolics Content in Different Varieties

Previous study also revealed the total phenolic content of five Philippine sweet potato varieties ranging from 50.1 to 362.8 mg GAE/100 g FW, which was quite similar to the results of our experiment [16]. Plenty of previous studies demonstrated that purple-fleshed varieties had higher total phenolic content than the yellow or white ones. The similar regulation was also found in our study [3,5].

### 3.2. Comparison of Phytochemical Composition and PCA Analysis of Ten Varieties

All of the polar phytochemical compounds or their derivatives detected in our study were also reported by prior studies, ensuring the reliability of our results. However, the content of these metabolites was quite different from the prior studies mainly because of the different extraction conditions and the HPLC method [3]. According to the results, we presumed that ferulic acid and caffeic acid could be the prominent phenolic acids and hyperoside was the prominent flavonoid in SPSRs. Those three compounds could be indicators for estimating the nutritional quality of SPSPs.

However, anthocyanin—another group of bioactive compounds contained in purple-fleshed SPSRs with strong antioxidant activity—was not calculated in this investigation unlike many previous studies [5,12,17]. Contrary to Aimin Wang’s work, the composition of three purple-fleshed varieties selected in our study was quite different based on the PCA analysis [3]. We assumed that it might be the result of the incompletely identification of phytochemical compounds in our study. Further study will focus on the group of anthocyanin in SPSRs especially the purple-fleshed varieties to get better comprehension of the phytochemical constitution of SPSRs, which could provide more accurate guidance for widening utilization of SPSRs in health-related area.

The PCA analysis revealed the correlation among the samples and metabolites detected in our study, which might help with developing biofortification strategies. Heperoside, ferulic acid and caffeic acid were prominent among those metabolites detected in our study with many health-related benefits, which could be set as a biofortification target in SPSRs. Hyperoside was found not related to epicatechin, isorhamnetin, isoquercetin and syringic acid but had positive correlation with caffeic acid. The mechanism behind this required a further research. A fortification of hyperoside might attribute to a higher content of caffeic acid so that the quality of could be enhanced due to the synergistic effect.

### 3.3. Comparison of Total Antioxidant Activity

From the highest to the lowest antioxidant activity the order is YS43, YZ7, YY153, YS25, HX22, XY34, WS7, YS15, YS7 and CS1, which was similar to the order of total phenolic content shown in Figure 2. It is reasonable to predict that there could be some kind of association between phenolic content and antioxidant activities. In fact, the following correlation analysis told a distinct positive relevancy between antioxidant activity estimated by ORAC assay and the total phenolic content. The similar trend was also reported by previous research, thus phenolics made great contribution to antioxidant capacity [16,18,19]. Limited research could be found about the ORAC values of SPSRs. The work completed by Jace D. Everette and Shahidul Islam in 2012 reported the ORAC values of orange-fleshed SRPs ranged from 23.44 to 45.51 mmol TE/100 g DW, equalling 6.45–12.52 mmol TE/100 g FW based on the moisture content mentioned in the material part [20]. Unexpectedly, probably due to the variety differences, all the ORAC values except YS43 were below this scope. Carrying out other assays such as ABTS, DPPH for in vitro antioxidant activity measurement might be effective to get more comprehensive understanding of in vitro antioxidant activity of SPSRs.

### 3.4. Comparison of Cellular Antioxidant Activity

Unexpectedly, among Table 2, the varieties with higher ORAC values especially the purple-fleshed varieties including YS43, YZ7 and YY153 were not all performed well in CAA test. We speculated there was no distinct relevance between the fleshed colour and the cellular antioxidant activities so that flesh colour could not be used as an indicator for cellular antioxidant activities.

Extract of YS43 had the strongest total antioxidant activities among ten varieties selected in our study. However, its antioxidant activities may not act on HepG2 cells since the CAA values of YS43 made numbered contribution to total antioxidant activities. We assumed that the extracts with similar phytochemical composition like YS25 and YS43 was able to well across the membrane of HepG2 cell and kept intracellular antioxidant activities. This hypothesis required more proof from further research. On the contrary, extract of CS1 could barely go through the cell membrane of HepG2. Likewise, extracts of HX22, YS7 and YS15 who had similar phytochemical composition according to PCA analysis shown in Figure 3 also had lower CAA values among ten varieties at the range of 0 ~ 30.15 ± 5.87 μmol QE/100 g FW. This may provide an indication for revealing the association between the phytochemical composition and the cellular antioxidant activity.

Limited research about the CAA test of SPSRs could be found. An impressive work conducted by Wei Song in 2010 reported that the CAA values with PBS wash was 1.78 ± 0.16 μmol QE/100 g FW [21]. We used the similar vegetable extraction method as well as the CAA assay of this study, but the CAA values of our study provided a broader coverage of 0.46 ± 0.08 ~ 181.15 ± 18.24 μmol QE/100 g FW in PBS wash protocol, covering more varieties and supplying a helpful reference for further investigations.

### 3.5. Comparison of Antiproliferative Activity against Human Liver Cancer HepG2 Cell

We indicated that the high activeness of antiproliferation of YS43 might be the result of both its high phenolic content and cell membrane transmittance properties which had been presented above. Previous studies have only disclosed the antiproliferative or anticancer activities of SPSRs extracts against NB4, MCF-7, HCT-116, HeLa, SNU-1, WiDr and MDBK cell lines, mostly presenting as EC_50_ values as well as line chart [11,12,13,14,19]. To the best of our knowledge, we were the first one who gave the antiproliferation test of SPSRs extracts to HepG2 cell line. The similar dose-dependent manner was also observed in other tumour cell lines antiproliferation studies mentioned above. The results of our studies could well fill the vacancy and give more comprehensive information about the antiproliferation in cancer cells for indicating the anticancer effect of SPSRs.

## 4. Materials and Methods

### 4.1. Sweet Potato Samples Preparation

Ten varieties of sweet potato in different colours from white to yellow-orange to dark purple including: YS7, HX22, YS43, WS7, YS25, YZ7, YY153, CS1, XY34 and YS15 were identified and provided by Chunxian Yang (Southwest University, Chongqing, China). Fresh storage roots were collected in the harvesting period and washed carefully with clean water to remove the muds and clays on the surface. All the samples were cut to small pieces and kept at −80 °C for analysis. Considering of all the following results were presented in fresh weigh, it was necessary to mention that the moisture content of SPSRs ranged from 62.8 to 82.2 g water/100 g FW, at the average of 72.5 g water/100 g FW, according to previous studies [2].

### 4.2. Phenolic Extraction for Sweet Potato

The method of phenolic extraction from storage roots of sweet potato was used as reported previously [22]. Briefly, 50 g fresh samples were extracted with 200 mL chilling 80% acetone for three times. All the extracted solutions were collected with vacuum filtration and evaporated to less than 10% initial volume at 45 °C. The phenolic extracts were redissolved with distilled water to a 50 mL volumetric flask. All the extractions were performed in triplicate and stored at −80 °C before analysis.

### 4.3. Determination of Phenolics for Sweet Potato Extracts

The phenolic content was measured by colourimetric Folin–Ciocalteu method as reported previously [23]. Folin–Ciocalteu reagent was used as dye reactive solution, and sodium carbonate (7%) was used to provide alkaline environment. Gallic acid was used as standard to calculate total phenolic content. Determine wavelength was set at 760 nm with a spectrophotometer. The data was expressed as milligram gallic acid equivalents per 100 g in fresh weight (mg GAE/100 g FW). All experiments were repeated three times and reported as mean ± SD for three replicates.

### 4.4. Phenolic Profiles of Sweet Potato Extracts Analysis by HPLC-PAD

Phenolic profiles were analysed with a Waters HPLC system (Waters Corporation, Milford, MA, USA) and the procedure was reported in our lab previously [24]. Waters XSelect HSS C18 column (100Å, 5 µm, 4.6 mm × 150 mm) was used to separate different compounds in sweet potato extracts. Determine wavelengths were set at 280 nm and 324 nm in PAD (Waters Corporation, Milford, MA, USA) detector for phenolic acids and flavonoids analysis. The sweet potato extracts were identified by different retention time with a gradient system (A: distilled water with 0.1% trifluoroacetic acid; B: methanol): 0–20 min 20% B, 20–25 min 25% B, 25–40 min 30% B, 40–60 min 40%, 60–75 min 50% B, 75–80 min 70% B, 80–85 min 20% B, 85–90 min 20% B. Chromatographic peaks were identified by comparing the retention time in specific wavelength spectra with those of authentic standards. Data was reported as mean ± SD (*n* = 3).

### 4.5. Total Antioxidant Activity Analysis for Sweet Potato Extracts

The antioxidant activities were conducted by oxygen radical absorbance capacity (ORAC) assay as described previously [25,26]. Trolox was used as standard, and fluorescein was used as fluorescence probe for analysis. Fluorescence intensity was measured at excitation of 485 nm and emission of 535 nm by Microplate Reader. The ORAC value of total antioxidant activity was expressed as mmol Trolox equivalents per 100 g in fresh weight (mmol TE/100 g FW). Data was reported as mean ± SD (*n* = 3).

### 4.6. Cellular Antioxidant Activity Analysis for Sweet Potato Extracts

The CAA assay was conducted as described previously [15,27]. Human live cancer cell line (HepG2, ATCC HB-8065) was used as cellular model in this assay; quercetin was used as standard to calculate the CAA value. HepG2 cells were seeded at a density of 6 × 10^4^ cells/well on a 96-well microplate and exposed to extracts for one hour for antioxidant activity analysis. The tested doses ranged from 2.5 mg FW/mL to 200 mg FW/mL. DCFH-DA was used as fluorescence probe and ABAP was used as free radical donor. Only PBS wash treatments were used in this assay. Fluorescence intensity was measured at excitation of 485 nm and emission of 535 nm for a dynamic analysis by Microplate Reader. CAA value was calculated from the integrated area under the fluorescence versus time curve, and the results were expressed as micromoles of quercetin equivalents (QE) per 100 g in fresh weight (μmol QE/100 g FW).

### 4.7. Determination of Antiproliferative and Cytotoxicity Activities of Sweet Potato Extracts

The cytotoxic activity and antiproliferative effects of sweet potato extracts were detected with the methods as described previously [28,29,30]. HepG2 cells were seeded at a density of 2.5 × 10^4^ cells/well on a 96-well microplate for antiproliferation activity analysis and 4 × 10^4^ cells/well for cytotoxicity analysis. Methylene blue was used as dye solution in the two assays. [28] The absorbance was measured at 570 nm by microplate reader. The cytotoxic activity was assessed by the half lethal dose (CC_50_), and the antiproliferative activity was assessed by the half maximal inhibitory concentration (IC_50_), which were expressed as mean ± SD mg FW/mL (*n* = 3).

### 4.8. Statistical Analysis

The evaluation of statistical significance of observed differences was performed by one-way analysis of variance (One-way ANOVA), the differences between means were determined using Duncan’s multiple comparison test, and the statistical significance was set at *p* < 0.05 using SPSS software 21.0 (SPSS Inc., Chicago, IL, USA).

## 5. Conclusions

Comparatively, the purple-flesh varieties, including YS43, YZ7 and YY153, who had higher phenolics content, also had corresponding stronger total antioxidant activities and antiproliferative activities against HepG2 cells. The significant correlation between phenolic content and total antioxidant activity was also determined by Pearson correlation analysis. (*p* < 0.05) Ferulic acid and caffeic acid could be the prominent phenolic acids and hyperoside was the prominent flavonoid in SPSRs. The relevance revealed by correlation analysis among those three prominent and other compounds provided new strategies for biofortification in SPSRs. HX22, YS15 as well as YS7, with a similar phytochemical composition according to PCA analysis, caused lower intracellular antioxidant capacity. However, purple-fleshed varieties mentioned above with abundant phenolics content did not do well in CAA test. We assumed that intracellular antioxidant capacity in HepG2 cell line was not related to total phenolic content and the flesh colour. Limited research about the intracellular antioxidant activity could be found. Our research can well fill the vacancy and provide information for design an appropriate diet from the perspective of antioxidant value level. Additionally, we are the first to perform the antiproliferation test of SPSRs extracts against HepG2 cell, and the results proved that SPSRs extracts did have anticancer effect at certain concentration, widening the utilization of sweet potatoes. Information provided by our study can bring a more comprehensive understanding of SPSRs for further utilization in health-related area.

## Figures and Tables

**Figure 1 molecules-24-04476-f001:**
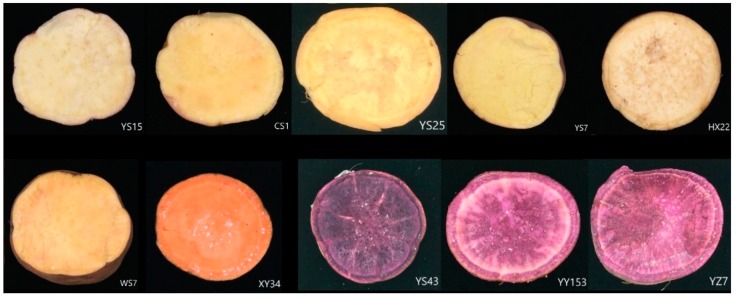
Cross section of the storage roots of the sweet potato varieties selected for the study.

**Figure 2 molecules-24-04476-f002:**
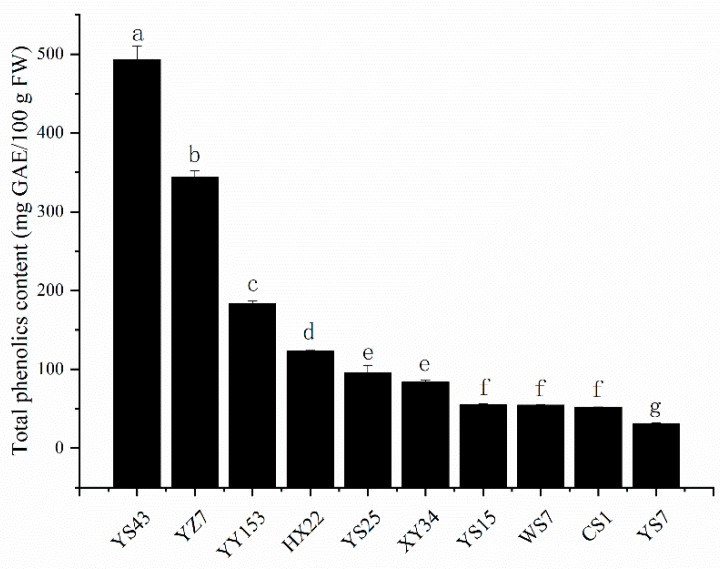
Comparison of total phenolics content of ten varieties sweet potato roots. Bars with different letters are significantly different (Duncan test, *p* < 0.05).

**Figure 3 molecules-24-04476-f003:**
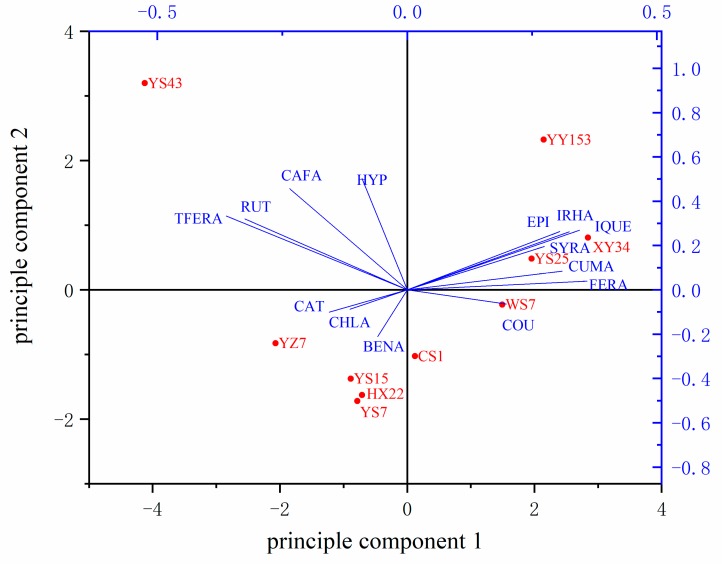
Principle component analysis of ten varieties sweet potato roots. RUT: rutin hydrate; COU: coumarin; EPI: epicatechin; CAT: catechin; HYP: hyperoside; IRHA: isorhamnetin; BENA: benzoic acid; IQUE: isoquercetin; TFERA: trans-ferulic acid; FERA: ferulic acid; CUMA: cumaric acid; SYA: syringic acid; CAFA: caffeic acid; CHLA: chlorogenic acid.

**Figure 4 molecules-24-04476-f004:**
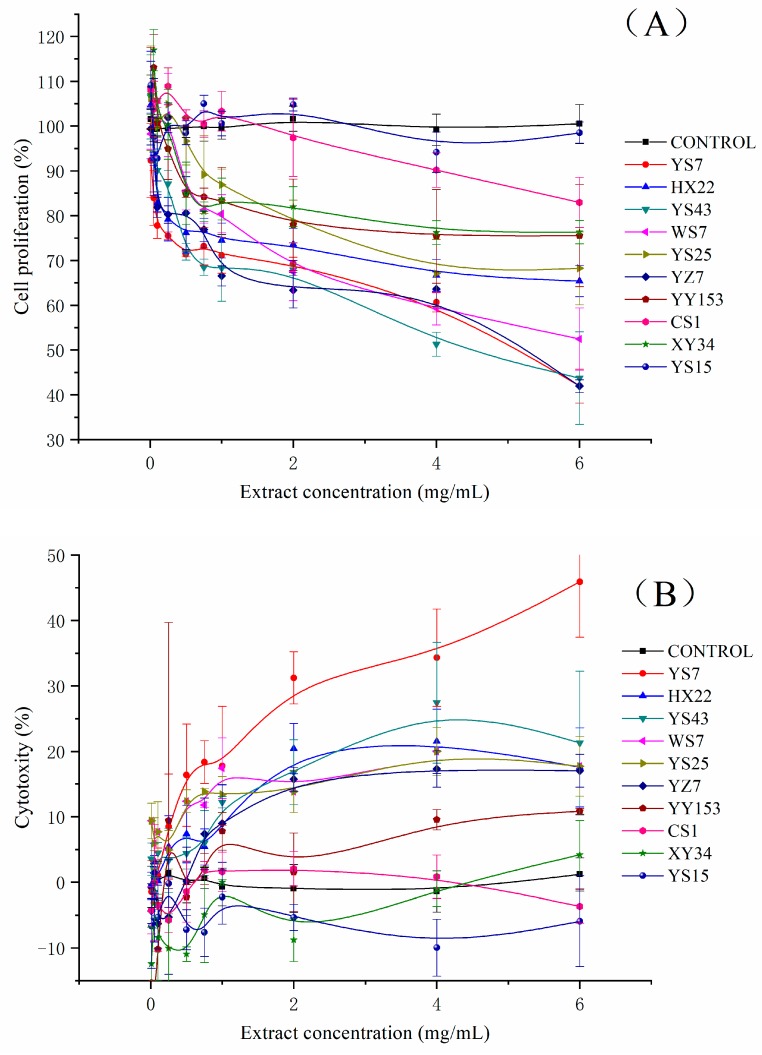
(**A**) Antiproliferative and (**B**) cytotoxic activities of the extracts against HepG2 cells.

**Figure 5 molecules-24-04476-f005:**
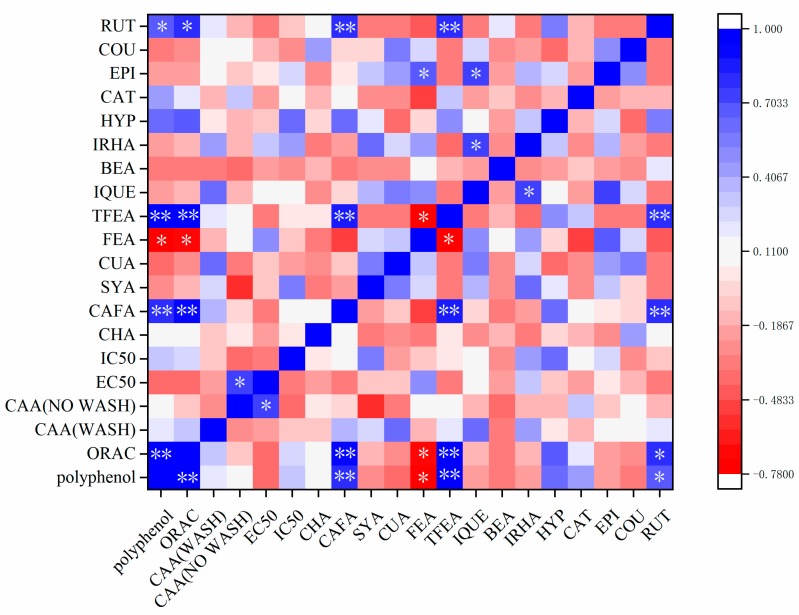
Pearson correlation analysis among phytochemical compounds, antioxidant activities, antiproliferative activities and cytotoxicity. * and ** mean significant correlation at the 0.05 and 0.01 level, respectively (2-tailed). RUT: rutin hydrate; COU: coumarin; EPI: epicatechin; CAT: catechin; HYP: hyperoside; IRHA: isorhamnetin; BEA: benzoic acid; IQUE: isoquercetin; TFEA: trans-ferulic acid; FEA: ferulic acid; CUA: cumaric acid; SYA: syringic acid; CAFA: caffeic acid; CHA: chlorogenic acid.

**Table 1 molecules-24-04476-t001:** Comparison of phytochemical composition of ten varieties sweet potato storage roots with HPLC.

Identified Compound	YS7	HX22	YS43	WS7	YS25	YZ7	YY153	CS1	XY34	YS15
chlorogenic acid	ND	20.11 ± 1.68a	5.34 ± 0.32b	0.51 ± 0.01c	0.67 ± 0.01c	ND	ND	ND	1.37 ± 0.09c	ND
caffeic acid	ND	0.56 ± 0.002cd	4.57 ± 1.18a	0.96 ± 0.08bc	0.59 ± 0.05cd	1.39 ± 0.001b	0.93 ± 0.01bc	0.13 ± 0.01d	1.55 ± 0.01b	0.10 ± 0.01d
syringic acid	ND	ND	ND	0.73 ± 0.11a	0.74 ± 0.13a	ND	0.67 ± 0.02a	ND	ND	ND
cumaric acid	ND	ND	ND	1.48 ± 0.08a	1.46 ± 0.02a	ND	ND	ND	1.46 ± 0.01a	ND
ferulic acid	19.13 ± 0.24d	8.25 ± 0.05g	ND	18.62 ± 0.01e	10.86 ± 0.45f	ND	26.03 ± 0.28b	23.98 ± 0.04c	27.97 ± 0.13a	1.31 ± 0.003h
trans-ferulic acid	ND	ND	1.34 ± 0.03a	ND	ND	0.58 ± 0.02b	ND	ND	ND	ND
isoquercetin	ND	ND	ND	ND	0.43 ± 0.03b	ND	0.41 ± 0.01c	ND	0.60 ± 0.01a	ND
benzoic acid	12.37 ± 2.60	ND	ND	ND	ND	ND	ND	ND	ND	ND
isorhamnetin	ND	ND	ND	ND	2.13 ± 0.19b	ND	2.25 ± 0.02a	0.81 ± 0.03c	0.87 ± 0.04c	ND
hyperoside	ND	ND	417.6 ± 14.1b	7.94 ± 0.97e	ND	14.78 ± 1.75cd	471.3 ± 2.5a	16.34 ± 1.10c	11.13 ± 1.24cde	9.26 ± 0.89de
catechin	ND	ND	ND	ND	ND	2.15 ± 0.17a	ND	ND	ND	ND
epicatechin	ND	ND	ND	0.17 ± 0.001c	ND	ND	0.30 ± 0.004b	ND	0.35 ± 0.02a	ND
coumarin	ND	0.26 ± 0.01b	ND	0.23 ± 0.001b	0.03 ± 0.01d	0.04 ± 0.004c	ND	ND	0.34 ± 0.02a	ND
rutin hydrate	10.53 ± 2.51b	ND	33.38 ± 3.09a	ND	ND	ND	ND	ND	ND	ND

Unit: mg/100 g FW; Different letters in the same row indicate significant differences (Duncan test, *p* < 0.05).

**Table 2 molecules-24-04476-t002:** Antioxidant activities of ten varieties sweet potato storage roots.

	Total Antioxidant Activities (mmol TE/100 g FW)	Cellular Antioxidant Activities (μmol QE/100 g FW)	Contribution (%)	Cyctoxicity (mg/mL)
Varieties	ORAC	PBS Wash	PBS Wash/ORAC	CC_10_
YS7	0.45 ± 0.07^d^	3.34 ± 0.48^b^	0.74	0.698
HX22	1.73 ± 0.03^cd^	30.15 ± 5.87^b^	1.74	1.993
YS43	7.38 ± 0.98^a^	120.68 ± 36.9^a^	1.64	1.662
WS7	1.17 ± 0.18^d^	38.47 ± 8.57^b^	3.29	1.977
YS25	1.90 ± 0.28^cd^	23.14 ± 3.86^b^	1.22	2.013
YZ7	3.39 ± 0.36^b^	50.73 ± 7.56^b^	1.50	2.478
YY153	2.38 ± 0.18^bc^	35.22 ± 6.97^b^	1.48	5.304
CS1	0.40 ± 0.03^d^	0.46 ± 0.08^b^	0.12	ND
XY34	1.19 ± 0.15^d^	181.15 ± 18.24^a^	15.22	ND
YS15	0.52 ± 0.04^d^	ND	ND	ND

Different letters in the same column indicate significant differences (Duncan test, *p* < 0.05); ND= not detected.

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
