# Peer review of "Comparative Assessment of Phenolic Profiles, Cellular Antioxidant and Antiproliferative Activities in Ten Varieties of Sweet Potato (Ipomoea Batatas) Storage Roots"

_molecules, 2019, doi:10.3390/molecules24244476_

Round 1
Reviewer 1 Report
The paper provides a comparative assessment of phenolic profiles, cellular antioxidant and antiproliferative activities in ten varieties of sweet potato roots for which little prior information has been reported.
However, the study suggests providing a breeding strategy for bioactive compounds fortification in sweet potato as a functional crop and value-added food products to promote human health. Such research should be proved useful in the determination of the bioactivity to justify a broader use of sweet potato in health-related applications.
This is a speculative leap and needs to be addressed in the article because antioxidant capacity is not related to human health. Consuming phytochemicals which are antioxidants in the laboratory is now well regarded as unlikely to be the mechanism by which human health is enhanced following fruit phytochemical consumption. The marketing of many food products has yet to catch up with the science.
The authors of the manuscript need enlightening substantially the results presented. Several clarifications are required for contribute with the manuscript, as follow:
Stat of art, of the topic developed in this work, as well as the novelty and the contribution to knowledge, should be highlighted and included in the Introduction.
The authors should be deepening the discussion about the results obtained in this manuscript.
The quality of study (scientific novelty) could be improve if the results observed being compared with other studies.
Authors measured the cellular antioxidant activity using two different protocols. What is the rationale of using two different protocols? Do they provide different information? In my opinion only the protocol that include cell washing with PBS is appropriate as it removes the excess of the fluorescent probe (DCFH-DA) that has not penetrated the cells. My suggestion is to present in Table 2 only the data of CAA based on PBS wash protocol and rewritten CAA results and discussion section accordingly.
The conclusion of the study presented does not reflect the experimental findings.
Author Response
Dear Reviewer,
Thank you much for your comments with our manuscript. We have checked and revised the manuscript according to your comments as following:
About antioxidant capacity and consuming phytochemical compounds.Actually, we also agree that consuming the same composition of phytochemical compounds in fruit or vegetable might not provide same benefits for human health. While, investigating further information of the bioactive compounds might be helpful for determining the main functional components in sweet potato roots. The issue about the relevance between human health and antioxidant capacity is still controversial. But we believe that antioxidant capacity could be an implication or indicator for human diet and health and also for food preservation. Epidemiological and clinical studies show that diets with a high antioxidant capacity are related to significant decreases in plasma C-reactive protein, ischemic stroke or the overall risk of cardiovascular disease and colorectal cancer. Diets with appropriate antioxidant values might have some benefits for human health. Measuring the antioxidant capacity of sweet potato root might provide more information for designing a healthier diet.
Introduction, conclusion and discussion parts have been carefully modified. Please let me know if there are still questions required to be addressed.
Thanks for your suggestion for the CAA assay. This assay does have a glaring omission, resulting in an inaccurately estimate about the antioxidant capacity in no PBS wash protocol. We have checked and modified the CAA part according to your valuable suggestions.
Reviewer 2 Report
The article reports the results of a comparative study carried out on the 80% acetone extracts obtained from the storage roots of ten cultivated varieties of sweet potato (Ipomoea batatas (L.) Lam., Convolvulaceae). The study includes the quali-quantitative characterization of the phenolic compounds contained in the extracts and the evaluation of the in vitro antioxidant capacity in a cell-free system (ORAC assay) and in human liver cancer cells (HepG2). Besides, the cytotoxic and antiproliferative activities of the extracts in HepG2 cells were investigated for the first time. The results highlighted that the extracts from purple-fleshed varieties have greater total phenolic content than the others, also exhibiting stronger total antioxidant and antiproliferative activities.
Originality/Novelty: The paper addresses a very interesting topic. Due to the economic relevance as food resource widely cultivated and consumed throughout the world, and considering that it is a valuable source of several nutraceutical components, sweet potato is now being recognized as a functional food. Indeed, a number of researches on the phytochemical composition and the biological properties of sweet potato have been carried out in the last decades. In my opinion the article is original, since it provides further information for a breeding program aimed at the enhancement of bioactive compounds, also increasing the knowledge of the biological properties of sweet potato storage roots.
Quality of presentation: The abstract describes the essential information of the work. The experimental design is appropriate, the methods used and the results are described quite adequately (only a few missing information should be provided) and supported by suitable figures and tables. Nonetheless, some improvements are needed. The Discussion paragraph is divided into separate sections, making it not very effective. I would suggest to unify Results and Discussion in a single paragraph. References should be numbered in order of appearance in the text and listed in the same order (not in alphabetical order) at the end of the manuscript. Besides, additional references should be provided to consolidate a few statements, whereas some references are not appropriate and should be replaced. The English forms of some sentences are not clearly understandable and should be revised, and various typing and grammatical inaccuracies need to be addressed (for example line 45 “abundance”, line 53 “there is still”, line 58 “food product”, line 68 “one”, etc.). The authors should carefully check the manuscript.
Listed below some suggested modifications are reported:
Introduction
- Line 32: The Authors should indicate the correct botanical name of the plant: Ipomoea batatas (L.) Lam., family Convolvulaceae.
- Line 41: The sentence “Phenolics are the secondary metabolites for biotic stress and abiotic stress in higher plants” is unclear. It could be changed into “Phenolics are secondary metabolites whose synthesis and accumulation is generally stimulated in response to biotic or abiotic stresses in higher plants”.
- Lines 44-46: The sentence “Plant-based foods, such as fruits, vegetables, potatoes, grains and legumes, which contain abundant of phenolics to prevent chronic diseases and to provide desirable health benefits” is unclear. It could be changed into “Plant-based foods, such as fruits, vegetables, potatoes, grains and legumes, contain phenolic compounds which have been shown to play a role in the prevention of chronic diseases and to provide desirable health benefits”.
- Lines 49-53: Bibliographic references supporting these statements should be added.
Results
- Lines 75-76: The sentence “There are total fourteen phytochemical compounds detected in our study, mostly were flavonoid and phenolic acid, showing in Table 1 with RP-HPLC method” is unclear.
- Line 108: The sentence “With a PBS wash protocol as well as a no PBS wash protocol after treated by samples” is unclear.
- Line 114: The sentence “To guaranteed the accuracy of the results, all the test concentration was controlled under CC10…” is unclear. It could be changed into: “To ensure the accuracy of the results, the extracts were tested at concentrations lower than CC10…”.
Discussion
- Lines 178-179: The sentence “However, another bioactive compound called anthocyanin which is also rich in purple-fleshed SPRs with strong antioxidant activities…” is unclear. It could be changed into “However, anthocyanins, another group of bioactive compounds contained in purple-fleshed SPRs with strong antioxidant activity…”. Furthermore, bibliographic references supporting this statement should be added, including Ref. 13 (Park et al., 2016). See for example:
Vishnu VR, et al. (2019). Comparative study on the chemical structure and in vitro antiproliferative activity of anthocyanins in purple root tubers and leaves of sweet potato (Ipomoea batatas). J. Agric. Food Chem. 67, 2467−2475.
Lee JH, et al. (2019). Intracellular Reactive Oxygen Species (ROS) removal and cytoprotection effects of sweet potatoes of various flesh colors and their polyphenols, including anthocyanin. Prev. Nutr. Food Sci. 24(3):293-298.
- Line 227: What do authors mean when they write “line char”? References 11 and 12 refer to Solanum species, they do not report studies on Ipomoea batatas. The sentence should be revised according to appropriate references concerning sweet potato roots. See for example:
Naz S, et al. (2017). Antioxidant, antimicrobial and antiproliferative activities of peel and pulp extracts of red and white varieties of Ipomoea batatas (L) Lam. Tropical Journal of Pharmaceutical Research 16 (9): 2221-2229.
Sugata M, et al. (2015). Anti-inflammatory and anticancer activities of Taiwanese purple-fleshed sweet potatoes (Ipomoea batatas L. Lam) extracts. BioMed Research International 2015:1-10.
Rabah IO, et al. (2004). Potential chemopreventive properties of extract from baked sweet potato (Ipomoea batatas Lam. cv. Koganesengan). J. Agric. Food Chem. 52(23):7152-7157.
Materials and Methods
- Lines 234-236: The voucher specimen numbers of the ten varieties should be indicated.
- Lines 237-238: The sentence “Fresh storage roots were collected in the harvested period and washed the surface with clean water….” is unclear. It could be changed into: “Fresh storage roots were collected in the harvesting period and washed carefully with water…”.
- Line 269: Are they μmol or mmol TE/100 g FW? (See Table 2).
- Line 285: The author should include in the experimental protocol the range of tested doses and the time of exposure of the cells to the extracts.
- Line 292: Is it Tukey's or Duncan’s multiple comparison test? See Lines 73, 93, and 104.
Figures and Tables
- Figure 1: The figure legend “Transverse photo of ten varieties sweet potato roots” is unclear. It could be replaced with: “Cross section of the storage roots of the sweet potato varieties selected for the study”.
- Figure 2: The figure legend “Duncan tests were carried out in each row and bars with different letters differ significantly at p < 0.05” is unclear. It could be replaced with: “Bars with different letters are significantly different (Duncan test, p < 0.05)”.
- Figure 4: The statistical differences are not indicated in the graphics. Further, the figure legend “(A) Antiproliferative activities and cytotoxicity of extracts against HepG2 cells. (B) cytotoxicity of extracts against HepG2 cells” could be replaced with “(A) Antiproliferative and (B) cytotoxic activities of the extracts against HepG2 cells”.
- Table 1: The sentence “Duncan tests were carried out in each compound with different letters differ significantly at p < 0.05” is unclear. It could be replaced with: “Different letters in the same row indicate significant differences (Duncan test, p < 0.05)”.
- Table 2: The sentence “Duncan tests were carried out in each compound with different letters differ significantly at p < 0.05” is unclear. It could be replaced with: “Different letters in the same column indicate significant differences (Duncan test, p < 0.05)”.
Minor points
- Line 64: It should be “Total Phenolic Content”.
- Line 66: “SPSRs” is the abbreviation given for sweet potato storage roots, whereas throughout the manuscript “SPRs” is reported.
- Line 78: It should be 0.13±0.01.
- Line 101: The authors wrote “CAA”, it should be “TE”.
- Line 135: The authors wrote“IC50 ranging from 5.162 mg/ml to 351 mg/ml”, but the lowest IC50 value reported for the extracts was of 4.663 mg/ml (Line 139).
- Line 146: “IC50” should be replaced with “CC50”.
- Line 177: “metabolisms” should be replaced with “metabolites”.
- Line 178: “extraction assay” could be replaced with “extraction conditions”.
- Lines 281-282: “cytotoxicity activity” should be changed into “cytotoxic activity”.
Author Response
Dear Reviewer,
Many thanks for your comments and suggestions for our manuscript. We have checked and revised the manuscript according to your comments specifically. Some spelling and grammar errors had been corrected as well. The new version of the manuscript was submitted. Hope these will make it more acceptable for publication. If you have any other question about this paper, please let me know without hesitation.
Some responses for your suggestion as following:
Line 32: The Authors should indicate the correct botanical name of the plant: Ipomoea batatas (L.) Lam., family Convolvulaceae.
Response: it was revised.
Line 41: The sentence “Phenolics are the secondary metabolites for biotic stress and abiotic stress in higher plants” is unclear. It could be changed into “Phenolics are secondary metabolites whose synthesis and accumulation is generally stimulated in response to biotic or abiotic stresses in higher plants”.
Response: it was revised.
Lines 44-46: The sentence “Plant-based foods, such as fruits, vegetables, potatoes, grains and legumes, which contain abundant of phenolics to prevent chronic diseases and to provide desirable health benefits” is unclear. It could be changed into “Plant-based foods, such as fruits, vegetables, potatoes, grains and legumes, contain phenolic compounds which have been shown to play a role in the prevention of chronic diseases and to provide desirable health benefits”.
Response: it was changed.
Lines 49-53: Bibliographic references supporting these statements should be added.
Response: it was added.
Lines 75-76: The sentence “There are total fourteen phytochemical compounds detected in our study, mostly were flavonoid and phenolic acid, showing in Table 1 with RP-HPLC method” is unclear.
Response: it was rewritten.
Line 108: The sentence “With a PBS wash protocol as well as a no PBS wash protocol after treated by samples” is unclear.
Response: it was rewritten.
Line 114: The sentence “To guaranteed the accuracy of the results, all the test concentration was controlled under CC10…” is unclear. It could be changed into: “To ensure the accuracy of the results, the extracts were tested at concentrations lower than CC10…”.
Response: it was changed.
Lines 178-179: The sentence “However, another bioactive compound called anthocyanin which is also rich in purple-fleshed SPRs with strong antioxidant activities…” is unclear. It could be changed into “However, anthocyanins, another group of bioactive compounds contained in purple-fleshed SPRs with strong antioxidant activity…”. Furthermore, bibliographic references supporting this statement should be added, including Ref. 13 (Park et al., 2016). See for example:
Vishnu VR, et al. (2019). Comparative study on the chemical structure and in vitro antiproliferative activity of anthocyanins in purple root tubers and leaves of sweet potato (Ipomoea batatas). J. Agric. Food Chem. 67, 2467−2475.
Lee JH, et al. (2019). Intracellular Reactive Oxygen Species (ROS) removal and cytoprotection effects of sweet potatoes of various flesh colors and their polyphenols, including anthocyanin. Prev. Nutr. Food Sci. 24(3):293-298.
Response: it was changed and the references were added.
- Line 227: What do authors mean when they write “line char”? References 11 and 12 refer to Solanum species, they do not report studies on Ipomoea batatas. The sentence should be revised according to appropriate references concerning sweet potato roots. See for example:
Naz S, et al. (2017). Antioxidant, antimicrobial and antiproliferative activities of peel and pulp extracts of red and white varieties of Ipomoea batatas (L) Lam. Tropical Journal of Pharmaceutical Research 16 (9): 2221-2229.
Sugata M, et al. (2015). Anti-inflammatory and anticancer activities of Taiwanese purple-fleshed sweet potatoes (Ipomoea batatas L. Lam) extracts. BioMed Research International 2015:1-10.
Rabah IO, et al. (2004). Potential chemopreventive properties of extract from baked sweet potato (Ipomoea batatas Lam. cv. Koganesengan). J. Agric. Food Chem. 52(23):7152-7157.
Response: it was changed and the references were added.
Lines 234-236: The voucher specimen numbers of the ten varieties should be indicated.
Response: it was revised in the manuscript.
Lines 237-238: The sentence “Fresh storage roots were collected in the harvested period and washed the surface with clean water….” is unclear. It could be changed into: “Fresh storage roots were collected in the harvesting period and washed carefully with water…”.
Response: it was changed.
Line 269: Are they μmol or mmol TE/100 g FW? (See Table 2).
Response: sorry, it was mistake. We have revised as mmol TE/100 g FW.
Line 285: The author should include in the experimental protocol the range of tested doses and the time of exposure of the cells to the extracts.
Response: it was revised.
Line 292: Is it Tukey's or Duncan’s multiple comparison test? See Lines 73, 93, and 104.
Response: it was revised.
Figure 1: The figure legend “Transverse photo of ten varieties sweet potato roots” is unclear. It could be replaced with: “Cross section of the storage roots of the sweet potato varieties selected for the study”.
Response: it was changed.
Figure 2: The figure legend “Duncan tests were carried out in each row and bars with different letters differ significantly at p < 0.05” is unclear. It could be replaced with: “Bars with different letters are significantly different (Duncan test, p < 0.05)”.
Response: it was changed.
Figure 4: The statistical differences are not indicated in the graphics. Further, the figure legend “(A) Antiproliferative activities and cytotoxicity of extracts against HepG2 cells. (B) cytotoxicity of extracts against HepG2 cells” could be replaced with “(A) Antiproliferative and (B) cytotoxic activities of the extracts against HepG2 cells”.
Response: it was changed.
Table 1: The sentence “Duncan tests were carried out in each compound with different letters differ significantly at p < 0.05” is unclear. It could be replaced with: “Different letters in the same row indicate significant differences (Duncan test, p < 0.05)”.
Response: it was changed.
Table 2: The sentence “Duncan tests were carried out in each compound with different letters differ significantly at p < 0.05” is unclear. It could be replaced with: “Different letters in the same column indicate significant differences (Duncan test, p < 0.05)”.
Response: it was changed.
Line 64: It should be “Total Phenolic Content”.
Response: it was revised.
Line 66: “SPSRs” is the abbreviation given for sweet potato storage roots, whereas throughout the manuscript “SPRs” is reported.
Response: it was revised.
Line 78: It should be 0.13±0.01.
Response: it was revised.
Line 101: The authors wrote “CAA”, it should be “TE”.
Response: it was revised.
Line 135: The authors wrote“IC50 ranging from 5.162 mg/ml to 351 mg/ml”, but the lowest IC50 value reported for the extracts was of 4.663 mg/ml (Line 139).
Response: it was revised.
Line 146: “IC50” should be replaced with “CC50”.
Response: it was revised.
Line 177: “metabolisms” should be replaced with “metabolites”.
Response: it was revised.
Line 178: “extraction assay” could be replaced with “extraction conditions”.
Response: it was revised.
Lines 281-282: “cytotoxicity activity” should be changed into “cytotoxic activity”.
Response: it was revised.
Reviewer 3 Report
Dear Authors
I have reviewed the manuscript " Comparative assessment of phenolic, cellular antioxidant and antiproliferative activities in ten varieties of sweet potatio storage roots" submitted to Molecules journal.
Sweet potato has high level of antioxidant activity and contains abundant of phenolics to prevent chronic disease in human body. The manuscript is provided a good information to compare ten varieties of sweet potato,
I have commented a several part in the following statement:
Line 4. Sceintific name will be Italic
Line 62. Not explain in any part (Figure 1). You may insert Figure 1 into the manuscript part
Sweet potato roots have been changed into sweet potato storage roots
Line 86, SPRs can be changed into SPSRs
Line 93. Table 1. rutin hydrate* -What is the meaning of * ?
Line 102. CAA--to write down full name (Total antixidant activities)
Line 102. mmol TE/100g FW----umol TE/100g FW
Line 118. Lower --- may be higher
Line 119. 3.34+0,48-------.046+0.08
Line 119, 120 ----umol QE/100g FW
Line 237. Spices can be changed pieces
Author Response
Dear Reviewer,
We are much appreciate the careful reading of our manuscript and valuable suggestions from you. All the comments and suggestions were considered carefully and addressed in the revised manuscript. Some of responses for your comments as following:
Line 4. Sceintific name will be Italic
Response: it was revised.
Line 62. Not explain in any part (Figure 1). You may insert Figure 1 into the manuscript part. Sweet potato roots have been changed into sweet potato storage roots
Response: it was revised.
Line 86, SPRs can be changed into SPSRs
Response: it was changed.
Line 93. Table 1. rutin hydrate* -What is the meaning of * ?
Response: it was revised.
Line 102. CAA--to write down full name (Total antixidant activities)
Response: it was revised.
Line 102. mmol TE/100g FW----umol TE/100g FW
Response: it was revised.
Line 118. Lower --- may be higher
Response: it was revised.
Line 119. 3.34+0,48-------.046+0.08
Response: it was revised.
Line 119, 120 ----umol QE/100g FW
Response: it was revised.
Line 237. Spices can be changed pieces
Response: it was revised.
Round 2
Reviewer 2 Report
I have no further comments.